# FlightBench: Benchmarking Learning-based Methods for Ego-vision-based Quadrotors Navigation

## ABSTRACT

Ego-vision-based navigation in cluttered environments is crucial for mobile systems, particularly agile quadrotors. While learning-based methods have shown promise recently, head-to-head comparisons with cutting-edge optimization-based approaches are scarce, leaving open the question of where and to what extent they truly excel. In this paper, we introduce FlightBench, the first comprehensive benchmark that implements various learning-based methods for ego-vision-based navigation and evaluates them against mainstream optimization-based baselines using a broad set of performance metrics. Additionally, we develop a suite of criteria to assess scenario difficulty and design test cases that span different levels of difficulty based on these criteria. Our results show that while learning-based methods excel in high-speed flight and faster inference, they struggle with challenging scenarios like sharp corners or view occlusion. Analytical experiments validate the correlation between our difficulty criteria and flight performance. We hope this benchmark and these criteria will drive future advancements in learning-based navigation for ego-vision quadrotors. The source code and documentation is available at `https://github.com/Anonymous314159265358/FlightBench`[1].

## 1 INTRODUCTION

Ego-vision-based navigation in cluttered environments is a fundamental capability for mobile systems and has been widely investigated (Anwar & Raychowdhury, 2018; Chi et al., 2018; Wang et al., 2020; Xiao et al., 2021; Stachowicz et al., 2023). It involves navigating a robot to a goal position without colliding with any obstacles in its environment, using equipped ego-vision cameras (Agarwal et al., 2023). Quadrotors, known for their agility and dynamism (Verbeke & Schutter, 2018; Loquercio et al., 2021), present unique challenges in achieving fast and safe flight. Traditionally, hierarchical methods address this problem by decoupling it into subtasks such as mapping, planning, and control (Xiao et al., 2024), optimizing the trajectory to avoid collisions. In contrast, recent works (Loquercio et al., 2021; Song et al., 2023; Kaufmann et al., 2023) have demonstrated that learning-based methods can unleash the full dynamic potential of agile platforms. These methods employ neural networks to generate a sequence of waypoints (Loquercio et al., 2021) or motion commands (Song et al., 2023; Kaufmann et al., 2023), utilizing state estimation and ego-vision input. Unlike the high computational costs associated with sequentially executed subtasks, this end-to-end manner significantly reduces processing latency and enhances agility (Loquercio et al., 2021).

Despite the promising results of learning-based navigation methods, the lack of head-to-head comparisons with state-of-the-art optimization-based methods makes it unclear in which areas they truly outperform and to what degree. Traditional methods are often evaluated using customized scenarios and sensor configurations (Ren et al., 2022; Zhou et al., 2019; Song et al., 2023), which complicates reproducibility and hinders fair comparisons. Moreover, the absence of a quantifiable approach for scenario difficulty further obscures the analysis of the strengths and weaknesses of current methods.

In this paper, we introduce FlightBench, a comprehensive benchmark that evaluates methods for ego-vision-based quadrotor navigation. We initially developed a suite of learning-based methods, encompassing both ego-vision and privileged ones for an in-depth comparative analysis. Additionally, we incorporated several representative optimization-based benchmarks for further head-to-head

---

[1]This is a temporary anonymous repository for double-blind review, which will be replaced upon acceptance.

Table 1: A comparison of FlightBench to other open-source benchmarks for navigation.

| Benchmark | 3D Scenarios | Classical Methods | Learning Methods | Sensory Input |
|---|---|---|---|---|
| MRBP 1.0 (Wen et al., 2021) | ✗ | ✓ | ✗ | LiDAR |
| Bench-MR (Heiden et al., 2021) | ✗ | ✓ | ✗ | - |
| PathBench (Toma et al., 2021) | ✓ | ✓ | ✓ | - |
| Gibson Bench (Xia et al., 2020) | ✗ | ✗ | ✓ | Vision |
| OMPLBench (Moll et al., 2015) | ✓ | ✓ | ✗ | - |
| RLNav (Xu et al., 2023) | ✗ | ✓ | ✓ | LiDAR |
| Plannie (Rocha & Vivaldini, 2022) | ✓ | ✓ | ✓ | - |
| **FlightBench (Ours)** | ✓ | ✓ | ✓ | Vision |

evaluation. Moreover, we established three criteria to measure the difficulty of scenarios, thereby creating a diverse array of tests that span a spectrum of difficulties. Finally, we compared these methods across a wide range of performance metrics to gain a deeper understanding of their specific attributes. Our experiments indicate that learning-based methods demonstrate superior performance in high-speed flight scenarios and generally offer quicker inference times. However, they encounter difficulties in handling complex situations such as sharp turns or occluded views. In contrast, our findings show that traditional optimization-based methods maintain a competitive edge. They perform well in challenging conditions, not just in success rate and flight quality, but also in computation time, particularly when they are meticulously designed. Our analytical experiments validate the effectiveness of the proposed criteria and emphasize the importance of latency randomization for learning-based methods. In summary, our key contributions include:

1. The development of FlightBench, the first unified open-source benchmark that facilitates the head-to-head comparison of learning-based and optimization-based methods on ego-vision-based quadrotor navigation under various 3D scenarios.

2. The proposition of tailored task difficulty and performance metrics, aiming to enable a thorough evaluation and in-depth analysis of specific attributes for different methods.

3. Detailed experimental analyses that demonstrate the comparative strengths and weaknesses of learning-based versus optimization-based methods, particularly in difficult scenarios.

## 2 RELATED WORK

**Planning methods for navigation.** Classical navigation algorithms typically use search or sampling to explore the configuration or state space and generate a free path (Kamon & Rivlin, 1997; Rajko & LaValle, 2001; Penicka & Scaramuzza, 2022). With optimization, a multi-objective optimization problem is often formulated to determine the optimal trajectory (Paull et al., 2012; Ye et al., 2022). This is commonly done using gradients from local maps, such as the Artificial Potential Field (APF) (Zhu et al., 2006; Sfeir et al., 2011) and the Euclidean Signed Distance Field (ESDF) (Zhou et al., 2019). On the other side, the development of deep learning enables algorithms to perform navigating directly from sensory inputs such as images or lidar (Xiao et al., 2024). The policies are trained by imitating expert demonstrations (Schilling et al., 2019) or through exploration under specific rewards (Liu et al., 2023). Learning-based algorithms have been applied to various mobile systems, such as quadrupedal robots (Agarwal et al., 2023), wheeled vehicles (Chaplot et al., 2020; Stachowicz et al., 2023), and quadrotors (Kaufmann et al., 2023; Xing et al., 2024). In this work, we examine representative methods for ego-vision-based navigation on quadrotors, including two learning-based approaches and three optimization-based methods, providing a comprehensive comparison between these categories.

**Benchmarks for navigation.** Several benchmarks exist for non-sensory-input navigation algorithms, such as OMPL (Moll et al., 2015), Bench-MR (Heiden et al., 2021), PathBench (Toma et al., 2021), and Plannie (Rocha & Vivaldini, 2022). OMPL and Bench-MR primarily focus on sampling-based methods, while PathBench evaluates graph-based and learning-based methods. Plannie offers sampling-based, heuristic, and learning-based methods for quadrotors. However, these algorithms do not utilize input from onboard sensors as well. For methods with sensory inputs, most

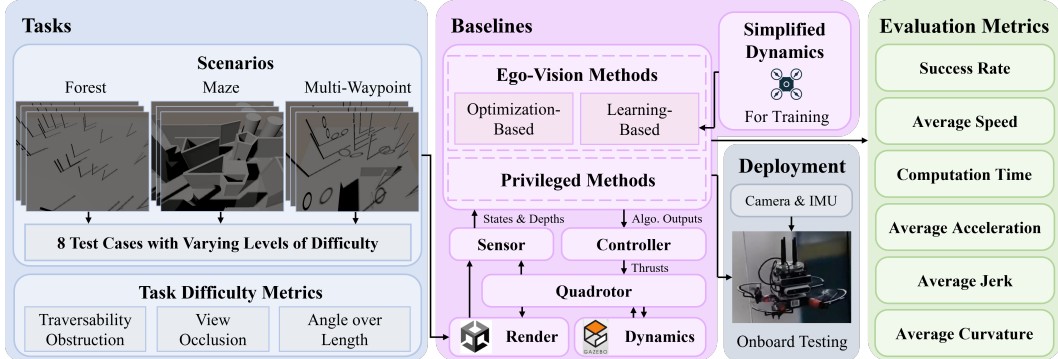

Figure 1: An overview of the FlightBench. FlightBench consists of three main components: (1) Tasks, featuring three scenarios categorized into eight difficulty levels. (2) Baselines, the core benchmarking platform supporting five ego-vision-based methods and two privileged methods. (3) Evaluation Metrics, offering a thorough suite of performance assessment metrics.

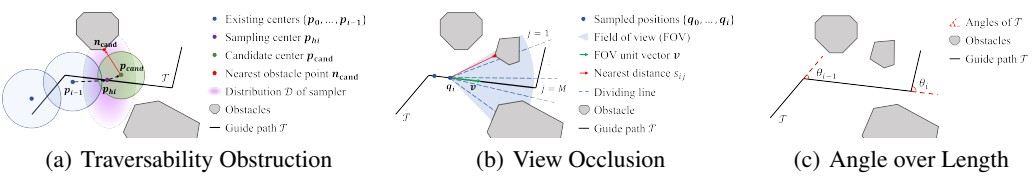

(a) Traversability Obstruction      (b) View Occlusion      (c) Angle over Length

Figure 2: Illustration of the task difficulty metrics

benchmarks are primarily designed for 2D scenarios. MRBP1.0 (Wen et al., 2021) and RLNav (Xu et al., 2023) evaluate planning methods using laser-scanning data for navigation around columns and cubes. GibsonBench (Xia et al., 2020) features a mobile agent equipped with a camera, navigating in interactive environments. As outlined in Tab. 1, there's a notable lack of a benchmark with 3D scenarios and ego-vision inputs to assess and compare both classical and learning-based navigation algorithms, a gap that FlightBench aims to fill.

## 3 FLIGHTBENCH

In this section, we detail the components of FlightBench. An overview of our benchmark is depicted in Fig. 1. In FlightBench, to design a set of `Tasks` with distinguishable characteristics for assessment, we propose three criteria, named task difficulty metrics, and develop eight tests across three scenarios based on these metrics. We integrate various representative `Baselines` to examine the strengths and features of both learning-based and optimization-based navigation methods. Furthermore, we establish a comprehensive set of performance `Evaluation Metrics` to facilitate quantitative comparisons. The next subsections will provide an in-depth look at the Tasks, Baselines, and Evaluation Metrics.

### 3.1 TASKS

#### 3.1.1 TASK DIFFICULTY METRICS

Each task difficulty metric quantifies the challenge of a test configuration from a specific perspective. In quadrotor navigation, a test is configured by the obstacles-laden scenario, start point, and end point. We utilize the topological guide path $\mathcal{T}$, which comprises interconnected individual waypoints (Penicka & Scaramuzza, 2022), to establish the quantitative assessment. In FlightBench, we propose three main task difficulty metrics: Traversability Obstruction (TO), View Occlusion (VO), and Angle Over Length (AOL).

**Traversability Obstruction.** Traversability Obstruction (TO) measures the flight difficulty due to limited traversable space caused by obstacles. We use a sampling-based approach (Ren et al., 2022) to construct sphere-shaped flight corridors $\{B_0, \ldots, B_{N_T}\}$, where $N_T$ is the number of spheres representing traversable space along path $\mathcal{T}$. Fig. 2(a) illustrates the primary notations for computing these corridors. The next sampling center $\mathbf{p}_{hi}$ is chosen from existing spheres $\{B_0, \ldots, B_{i-1}\}$ along $\mathcal{T}$. We sample $K$ candidate centers from a 3D Gaussian distribution $\mathcal{D}$ around $\mathbf{p}_{hi}$. Each candidate sphere $B_{cand}$ is defined by its center $\mathbf{p}_{cand}$ and radius $r_{cand} = ||\mathbf{p}_{cand} - \mathbf{n}_{cand}||_2 - r_d$, where $\mathbf{n}_{cand}$ is the nearest obstacle and $r_d$ is the drone radius. For each $B_{cand}$, we compute $S_{cand}$:

$$S_{cand} = k_1 V_{cand} + k_2 V_{inter} - k_3 (\mathbf{d} \cdot \mathbf{z}) - k_4 ||\mathbf{d} - (\mathbf{d} \cdot \mathbf{z})\mathbf{z}||_2 \tag{1}$$

where $k_1, k_2, k_3, k_4 \in \mathbb{R}^+$, $V_{cand}$ is the volume of $B_{cand}$, $V_{inter}$ is the overlap with $B_{i-1}$, $\mathbf{d} = \mathbf{p}_{cand} - \mathbf{p}_{hi}$, and $\mathbf{z}$ is the unit vector along $\mathbf{p}_{hi} - \mathbf{p}_i$. The sphere with the highest $S_{cand}$ is selected as the next sphere. This process repeats until path $\mathcal{T}$ is fully covered. Occlusion challenges mainly occur in narrow spaces, so sphere radii $\{r_1, \ldots, r_{N_T}\}$ are sorted in ascending order. The traversability obstruction metric $\mathbb{T}$ is defined in Eq. (2), where $R$ represents the sensing range.

$$\mathbb{T} = \frac{1}{N_T} \sum_{i=1}^{\lfloor N_T/2 \rfloor} \frac{R}{r_i}. \tag{2}$$

**View Occlusion.** In ego-vision-based navigation tasks, a narrow field of view (FOV) can limit the drone's perception (Tordesillas & How, 2022; Chen et al., 2024), posing a challenge to the perception capabilities of various methods (Gao et al., 2023). We use the term view occlusion (VO) to describe the extent to which obstacles block the FOV in a given scenario. The more obstructed the view, the higher the view occlusion. As shown in Fig. 2(b), we sample drone position $\{\mathbf{q_i}\}$ and FOV unit vector $\{\mathbf{v_i}\}$ along $\mathcal{T}$ with $i \in \{1, \cdots, N_V\}$. For each sampled pair $\{\mathbf{q_i}, \mathbf{v_i}\}$, we divide FOV into $M$ parts and calculate the distance $s_{ij}$ between the nearest obstacle point and drone position $q_i$ in each part $j$. The view occlusion $\mathbb{V}$ can be represented as Eq. (3), where $m_j$ is a series of weights, which gives higher weight to obstacles closer to the center of the view.

$$\mathbb{V} = \frac{1}{N_V} \sum_{i=1}^{N_V} \sum_{j=1}^{M} m_j \frac{R}{s_{ij}}. \tag{3}$$

**Angle Over Length.** For a given scenario, frequent and violent turns in traversable paths pose challenges for the agility of planning. Inspired by Heiden et al. (2021), we employ the concept of Angle Over Length (AOL) denoted as $\mathbb{A}$ to quantify the sharpness of a path. The AOL $\mathbb{A}$ is defined by Eq. (4), where $N_{AOL}$ signifies the number of angles depicted in Fig. 2(c), $\theta_i$ represents the $i$-th angle within the topological path $\mathcal{T}$, and $L$ stands for the length of $\mathcal{T}$.

$$\mathbb{A} = \frac{1}{L} \sum_{i=1}^{N_{AOL}} \left( \exp\left( \frac{\theta_i}{\pi/6} \right) - 1 \right). \tag{4}$$

### 3.1.2 SCENARIOS AND TESTS

As illustrated in Fig. 1, our benchmark incorporates specific tests based on three scenarios: `Forest`, `Maze`, and `Multi-Waypoint`. These scenarios were chosen for their representativeness and frequent use in evaluating the performance of quadrotor navigation methods. (Ren et al., 2022; Kaufmann et al., 2023). Within these scenarios, we developed eight tests, each characterized by varying levels of difficulty. The task difficulty scores for each test are detailed in Tab. 2.

The `Forest` scenario serves as the most common benchmark for quadrotor navigation. We differentiate task difficulty based on obstacle density and establish three tests, following the settings used

Table 2: Task difficulty score of each test case.

| Scenarios | Test Cases | TO | VO | AOL |
|-----------|-----------|------|------|------|
| Forest | 1 | 0.76 | 0.30 | $7.64 \times 10^{-4}$ |
| | 2 | 0.92 | 0.44 | $1.62 \times 10^{-3}$ |
| | 3 | 0.90 | 0.60 | $5.68 \times 10^{-3}$ |
| Maze | 1 | 1.42 | 0.51 | $1.36 \times 10^{-3}$ |
| | 2 | 1.51 | 1.01 | 0.010 |
| | 3 | 1.54 | 1.39 | 0.61 |
| MW | 1 | 1.81 | 0.55 | 0.08 |
| | 2 | 1.58 | 1.13 | 0.94 |

Table 3: Characteristics of the navigation methods for quadrotors. "RL" denotes reinforcement learning. "IL" represents imitation learning. "GM" and "EM" refer to Grid Mapping and ESDF Mapping, respectively. The control level indicates the part of the control stack used by the baseline.

| | Method Type | Priv. Info. | Decision Horizon | Mapping | Planning | Control Stack Traj. | Waypoint | Motion Cmd |
|---|---|---|---|---|---|---|---|---|
| SBMT | Samp.-based | ✓ | Global | | Planning Module | | | MPC |
| LMT | RL | ✓ | Local | | Policy Network | | | |
| Fast-Planner | Opti.-based | ✗ | Global | GM+EM | Planning Module | | | MPC |
| EGO-Planner | Opti.-based | ✗ | Local | GM | Planning Module | | | MPC |
| TGK-Planner | Opti.-based | ✗ | Global | GM | Planning Module | | | MPC |
| Agile | IL | ✗ | Local | | Policy Network | | | MPC |
| LPA | RL+IL | ✗ | Local | | Policy Network | | | |

by Loquercio et al. (2021). In the `Forest` scenario, TO and VO metrics increase with higher tree density. AOL is particularly low due to sparsely spanned obstacles, making this scenario suitable for high-speed flights (Loquercio et al., 2021; Ren et al., 2022).

The `Maze` scenario consists of walls and boxes, creating consecutive sharp turns and narrow gaps. Quadrotors must navigate these confined spaces while maintaining flight stability and perception awareness (Park et al., 2023). We devise three tests with varying lengths and turn complexities for `Maze`, resulting in discriminating difficulty levels for VO and AOL.

The `Multi-Waypoint` (`MW`) scenario involves flying through multiple waypoints at different heights sequentially (Song et al., 2021b). This scenario also includes boxes and walls as obstacles. We have created two tests with different waypoint configurations. The `MW` scenario is relatively challenging, featuring the highest TO in test 1 and the highest AOL in test 2.

### 3.2 BASELINES

Here, we introduce the representative methods for ego-vision-based navigation evaluated in Flight-Bench, covering two learning-based methods and three optimization-based methods, as well as two privileged methods that leverage access to environmental information. The characteristics of each method are detailed in Tab. 3. For links to the open-source code, key parameters, and implementation details, please refer to Appendix A.

**Learning-based Methods.** Utilizing techniques such as imitation learning (IL) and reinforcement learning (RL), learning-based methods train neural networks for end-to-end planning, bypassing the time-consuming mapping process. **Agile** (Loquercio et al., 2021) employs DAgger (Ross et al., 2011) to imitate an expert generating collision-free trajectories using Metropolis-Hastings sampling and outputs mid-level waypoints. **LPA** (Song et al., 2023) combines IL and RL, starting with training a teacher policy using **LMT** (Penicka et al., 2022), then distilling this expertise into a ego-vision-based student. Both teacher and student policies generate executable motion commands, specifically collective thrust and body rates (CTBR).

**Optimization-based Methods.** Optimization-based methods typically include an online mapping module followed by planning and control modules, often referred to as planners. The three optimization-based methods described below generate B-spline trajectories (see Tab. 3). The control stack samples mid-level waypoints from the trajectory function and uses a low-level Model Predictive Control (MPC) controller to convert these waypoints into motion commands (Fig. 3). Among these baselines, **Fast-Planner** (Zhou et al., 2019) constructs both occupancy grid and Euclidean Signed Distance Field (ESDF) maps, whereas **TGK-Planner** (Ye et al., 2020) and **EGO-Planner** (Zhou et al., 2020) only require an occupancy grid map. In the planning stage, **EGO-Planner** (Zhou et al., 2020) focuses on trajectory sections with new obstacles, acting as a local planner, while the other two use a global search front-end and an optimization back-end for long-horizon planning.

**Privileged Methods.** SBMT (Penicka & Scaramuzza, 2022) is a sampling-based method that uses global ESDF maps to generate a collision-free trajectory of dense points, which is then tracked by an MPC controller (Falanga et al., 2018). Its follow-up, **LMT** (Penicka et al., 2022), employs RL

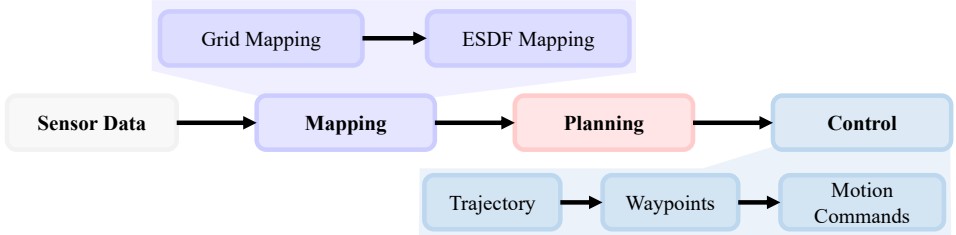

Figure 3: A generic processing pipeline for hierarchical navigation systems on quadrotors.

to train an end-to-end policy for minimum-time flight. This policy uses the quadrotor's full states and the next collision-free point (CFP) to produce motion commands, with the CFP determined by finding the farthest collision-free point on a reference trajectory (Penicka & Scaramuzza, 2022).

## 3.3 EVALUATION METRICS

We represent quadrotor states as a tuple $(\mathbf{x}(t), \mathbf{v}(t), \mathbf{a}(t), \mathbf{j}(t))$, where $t$ denotes time, $\mathbf{x}(t)$ denotes the position, and $\mathbf{v}(t) = \dot{\mathbf{x}}(t)$, $\mathbf{a}(t) = \dot{\mathbf{v}}(t)$, $\mathbf{j}(t) = \dot{\mathbf{a}}(t)$ are the velocity, acceleration, and jerk in the world frame, respectively. $T$ denotes the time taken to fly from the starting point to the end point.

First, we integrate three widely used metrics (Ren et al., 2022; Loquercio et al., 2021) into FlightBench. **Success rate** measures if the quadrotor reaches the goal within a 1.5 m radius without crashing. **Average speed**, defined as $\frac{1}{T}\int_0^T ||\mathbf{v}(t)||_2\, \mathrm{d}t$, reflects the achieved agility. **Computation time** evaluates real-time performance as the sum of processing times for mapping, planning, and control. Additionally, we introduce **average acceleration** and **average jerk** (Zhou et al., 2019; 2020), defined as $\bar{\mathbf{a}} = \frac{1}{L}\int_0^T ||\mathbf{a}(t)||_2^2\, \mathrm{d}t$ and $\bar{\mathbf{j}} = \frac{1}{L}\int_0^T ||\mathbf{j}(t)||_2^2\, \mathrm{d}t$, respectively. Average acceleration indicates energy consumption, while average jerk measures flight smoothness (Wang et al., 2022). These metrics, though not commonly used in evaluations, are crucial for assessing practicability and safety in real-world applications. However, average acceleration and jerk only capture the dynamic characteristics of a flight. For instance, higher flight speeds along the same trajectory result in greater average acceleration and jerk. To assess the static quality of trajectories, we propose **average curvature**, inspired by Heiden et al. (2021). Curvature is calculated as $\kappa(t) = \frac{|\mathbf{v}(t) \times \mathbf{a}(t)|}{|\mathbf{v}(t)|^3}$, and average curvature is defined as $\bar{\kappa} = \frac{1}{L}\int_0^T \kappa(t)\mathbf{v}(t)\, \mathrm{d}t$.

Together, these six metrics provide a comprehensive comparison of learning-based algorithms against optimization-based methods for ego-vision-based quadrotor navigation. Our extensive experiments will demonstrate that while learning-based methods excel in certain metrics, they also have shortcomings in others.

## 4 EXPERIMENTS

Upon FlightBench, which provides a variety of metrics and scenarios to evaluate the performance of ego-vision-based navigation methods for quadrotors, we carried out extensive experiments to study the following research questions:

- What are the main advantages and limitations of learning-based methods compared to optimization-based methods?
- How does the navigation performance vary across different scenario settings?
- How much does the system latency introduced by the practical evaluation environment affect performance?

## 4.1 SETUP

For simulating quadrotors, we use Flightmare (Song et al., 2021a) with Gazebo (Koenig & Howard, 2004) as its dynamic engine. To mimic real-world conditions, we develop a simulated quadrotor model equipped with an IMU sensor and a depth camera, calibrated with real flight data. The physical

Table 4: Performance evaluation of different methods under tests with the highest AOL within each scenario. The highest performing values for each metric are highlighted in bold, with the second highest underlined.

| Scen. | Metric | Privileged | | Optimization-based | | | Learning-based | |
|---|---|---|---|---|---|---|---|---|
| | | SBMT | LMT | TGK | Fast | EGO | Agile | LPA |
| Forest | Success Rate ↑ | 0.80 | **1.00** | 0.90 | 0.90 | **1.00** | 0.90 | **1.00** |
| | Avg. Spd. (ms$^{-1}$) ↑ | **15.25** | 11.84 | 2.30 | 2.47 | 2.49 | 3.058 | 8.96 |
| | Avg. Curv. (m$^{-1}$) ↓ | **0.06** | 0.07 | 0.08 | **0.06** | 0.08 | 0.37 | 0.08 |
| | Avg. Acc. (ms$^{-3}$) ↓ | 28.39 | 10.29 | 0.25 | **0.19** | 0.83 | 4.93 | 9.96 |
| | Avg. Jerk (ms$^{-5}$) ↓ | $4.27{\times}10^3$ | $8.14{\times}10^3$ | **1.03** | 3.97 | 58.39 | 937.02 | $1.14{\times}10^4$ |
| Maze | Success Rate ↑ | 0.60 | **0.9** | 0.50 | 0.60 | 0.20 | 0.50 | 0.30 |
| | Avg. Spd. (ms$^{-1}$) ↑ | 8.73 | **9.62** | 1.85 | 1.99 | 2.19 | 3.00 | 8.35 |
| | Avg. Curv. (m$^{-1}$) ↓ | 0.31 | **0.13** | 0.17 | 0.23 | 0.33 | 0.68 | 0.21 |
| | Avg. Acc. (ms$^{-3}$) ↓ | 60.73 | 26.26 | **0.50** | 0.79 | 1.91 | 15.45 | 37.30 |
| | Avg. Jerk (ms$^{-5}$) ↓ | $6.60{\times}10^3$ | $4.64{\times}10^3$ | **6.74** | 9.62 | 80.54 | $2.15{\times}10^3$ | $4.64{\times}10^3$ |
| MW | Success Rate ↑ | 0.70 | **0.90** | 0.40 | 0.80 | 0.50 | 0.60 | 0.50 |
| | Avg. Spd. (ms$^{-1}$) ↑ | 5.59 | **6.88** | 1.48 | 1.73 | 2.13 | 3.05 | 6.72 |
| | Avg. Curv. (m$^{-1}$) ↓ | 0.47 | 0.30 | 0.46 | 0.32 | 0.62 | 0.67 | **0.26** |
| | Avg. Acc. (ms$^{-3}$) ↓ | 80.95 | 31.23 | 1.07 | **0.97** | 5.06 | 16.86 | 36.77 |
| | Avg. Jerk (ms$^{-5}$) ↓ | $9.76{\times}10^3$ | $1.66{\times}10^4$ | 25.52 | **22.72** | 155.83 | $2.07{\times}10^3$ | $6.19{\times}10^3$ |

characteristics of the quadrotor and sensors are detailed in Appendix B. To simulate real-world communication delays, all data transmission in the simulation uses ROS (Quigley et al., 2009). All simulations are conducted on a desktop PC with an Intel Core i9-11900K processor and an Nvidia 3090 GPU. To evaluate real-time reaction performance on embedded platforms with limited computational resources, we also measure computation time on the Nvidia Jetson Orin NX module. Each evaluation metric is averaged over ten independent runs.

## 4.2 BENCHMARKING FLIGHT PERFORMANCE

### 4.2.1 FLIGHT QUALITY

To systematically assess the strengths and weaknesses of various methods on ego-vision navigation, we conduct evaluations across all tests within three distinct scenarios. Tab. 4 displays the results for tests with the highest AOL in each scenario. The comprehensive results for all tests are included in Appendix C.1 due to space limitations. We evaluate these methods using our proposed evaluation metrics, and computation time will be addressed separately in a subsequent discussion. In these experiments, we standardize the expected maximum speed at $3m/s$ for a fair comparison. Exceptions are SBMT, LMT, and LPA, whose flight speeds cannot be manually controlled.

As shown in Tab. 4, the privileged methods, with a global awareness of obstacles, set the upper bound for motion performance in terms of average speed and success rate. In contrast, the success rate of ego-vision methods in the Maze and MW scenarios is generally below $0.6$, indicating that our benchmark remains challenging for ego-vision methods, especially at the perception level.

Learning-based methods, known for their aggressive maneuvering, tend to fly less smoothly and consume more energy. They also experience more crashes in areas with large corners, as seen in the Maze and MW scenarios. When performing a large-angle turn, an aggressive policy is more likely to cause the quadrotor to lose balance and crash. Optimization-based methods are still competitive or even superior to current learning-based approaches, particularly in terms of minimizing energy costs. By contrasting the more effective Fast-Planner with the more severely impaired TGK-Planner and EGO-Planner, we find that global trajectory smoothing and enhancing the speed of replanning are crucial for improving success rates in complex scenarios. Detailed analyses on the failure cases can be found in the Appendix C.2.

**Remark.** *Learning-based methods tend to execute aggressive and fluctuating maneuvers, yet they struggle with instability in scenarios with high challenges related to VO and AOL.*

### 4.2.2 IMPACT OF FLIGHT SPEED

Table 5: Computation time of different baselines. $T_{\text{map}}$, $T_{\text{plan}}$, $T_{\text{ctrl}}$, $T_{\text{tot}}$ stands for mapping time, planning time, control time, and total time, respectively.

|  |  | SBMT | LMT | TGK | Fast | EGO | Agile | LPA |
|---|---|---|---|---|---|---|---|---|
| Desktop | $T_{\text{tot}}$ (ms) | $3.189 \times 10^5$ | 2.773 | 11.960 | 8.196 | 3.470 | 5.573 | 1.395 |
|  | $T_{\text{map}}$ (ms) | - | 1.607 | 3.964 | 7.038 | 2.956 | 0.338 | 0.399 |
|  | $T_{\text{plan}}$ (ms) | $2.589 \times 10^5$ | 1.167 | 7.994 | 1.155 | 0.510 | 5.115 | 0.995 |
|  | $T_{\text{ctrl}}$ (ms) | - | - | 0.002 | 0.003 | 0.003 | 0.119 | - |
| Onboard | $T_{\text{tot}}$ (ms) | - | 13.213 | 37.646 | 39.177 | 24.946 | 27.458 | 12.293 |
|  | $T_{\text{map}}$ (ms) | - | 2.768 | 27.420 | 36.853 | 24.020 | 1.175 | 4.313 |
|  | $T_{\text{plan}}$ (ms) | - | 10.445 | 10.211 | 2.310 | 0.910 | 26.283 | 7.980 |
|  | $T_{\text{ctrl}}$ (ms) | - | - | 0.015 | 0.014 | 0.016 | - | - |

As discussed above, learning-based methods, which tend to fly more aggressively with greater fluctuation, underperform in scenarios requiring large turning angles. This section examines the performance of various methods in the expansive Forest scenario as flight speeds vary. We conduct experiments under test 2 of the Forest scenario, which records the highest TO values. As shown in Fig. 4, we evaluate the success rate of each method at different average flight speeds, excluding three methods where speed is integral to the planning process and non-adjustable.

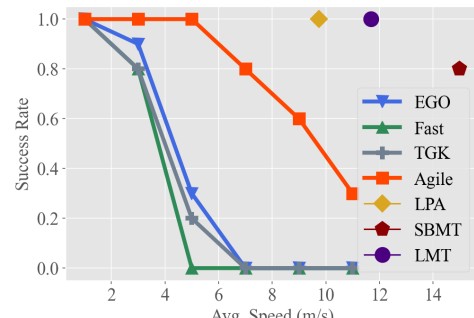

Figure 4: Success rate with different flight speeds.

Learning-based methods exhibit agile avoidance and operate closer to dynamic limits due to their straightforward end-to-end architecture. In contrast, optimization-based methods struggle in high-speed flights, as their hierarchical architecture and latency between sequential modules can cause the quadrotor to overshoot obstacles before a new path is planned. A detailed analysis of the computation time for the pipeline is discussed in Sec. 4.2.3. Additionally, privileged methods show higher success rates and faster flights, highlighting the substantial potential for improvement in current ego-vision-based methods.

**Remark.** *Learning-based methods consistently surpass optimization-based baselines for high-speed flight, yet they fall significantly short of the optimal solution.*

### 4.2.3 COMPUTATION TIME ANALYSES

To evaluate the computation time of various planning methods, we measured the time consumed at different stages on both desktop and onboard platforms. The results are presented in Tab. 5, with the computation time broken down into mapping, planning, and control stages as shown in Fig. 3. For learning-based methods, the mapping time includes converting images to tensors and pre-processing quadrotor states. Notably, SBMT optimizes the entire trajectory before execution, resulting in the highest computation time.

The computation time for learning-based methods is significantly influenced by neural network design. Agile exhibits longer planning time compared to LPA and LMT due to its utilization of a MobileNet-V3-small (Howard et al., 2019) encoder for processing depth images. Conversely, LPA and LMT employ simpler CNN and MLP structures, resulting in shorter inference time. Regarding optimization-based methods, the mapping stage often consumes the most time, particularly on the onboard platform. Fast-Planner experiences the longest mapping duration due to the necessity of constructing an ESDF map. EGO-Planner demonstrates superior planning time, even outperforming some end-to-end learning-based methods, owing to its local replanning approach. Nonetheless, this may pose challenges in scenarios with high VO and AOL. As discussed in Sec. 4.2.2, for algorithms requiring real-time onboard computation, computation time significantly influences the upper limit of flight speed. Algorithms with more lightweight architectures can operate at a higher replan frequency, enhancing their ability to react to sudden obstacles.

**Remark.** *Learning-based methods can achieve faster planning with compact neural network designs, while a well-crafted optimization-based methods can be equally competitive.*

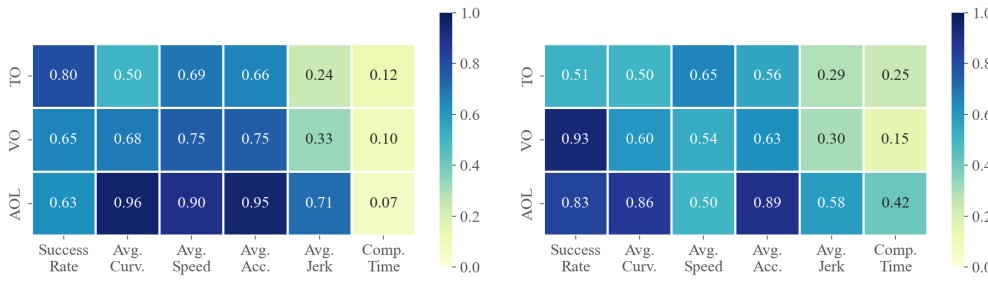

(a) Correlation heatmap for privileged methods    (b) Correlation heatmap for ego-vision methods

Figure 5: Correlation coefficients between three difficulty metrics and six evaluation metrics.

Table 6: Performance of RL-based methods under test 2 of Multi-Waypoint scenario.

|  | Train w/ latency | | Train w/o latency | |
|---|---|---|---|---|
|  | Success Rate ↑ | Progress | Success Rate ↑ | Progress |
| LMT | **0.9** | **0.96** | 0.3 | 0.57 |
| LPA | **0.5** | **0.73** | 0.0 | 0.32 |

### 4.3 ANALYSES ON EFFECTIVENESS OF DIFFERENT METRICS

To demonstrate how various task difficulties influence different aspects of flight performance, we calculate the correlation coefficients between six performance metrics and three difficulty metrics for each method across multiple tests. The value at the intersection of the horizontal and vertical axes represents the absolute value of the correlation coefficient between the two metrics. A higher value indicates a stronger correlation. Fig. 5 presents the average correlations for privileged and ego-vision-based methods, separately evaluating the impacts on agility and partial observation. Refer to Appendix C.3 for calculation details and the specific correlation coefficients for each method.

The results for privileged methods shown in Fig. 5(a) indicate that AOL and TO have a significant impact on the baseline's motion performance. The correlation coefficients between AOL and average curvature, velocity and acceleration are all above 0.9, indicating that AOL describes the sharpness of the trajectory well. More specifically, high AOL results in high curvature and acceleration of the flight trajectory, as well as lower average speed. TO, indicating task narrowness, is a crucial determinant of flight success rates. In contrast to privileged methods where global information is accessible, as shown in Fig. 5(b), ego-vision-based methods primarily struggle with partial perception. Field-of-view occlusions and turns challenge real-time environmental awareness, making VO and AOL highly correlated with success rates.

**Remark.** *High VO and AOL significantly challenge learning-based methods, as these factors heavily impact the ego-vision-based method's ability to handle partial observations and sudden reactions.*

### 4.4 IMPACT OF LATENCY ON LEARNING-BASED METHODS

Beyond the algorithmic factors previously discussed, learning-based methods face considerable challenges transitioning from training simulations to real-world applications. Latency significantly impacts sim-to-real transfer, particularly when simplified robot dynamics are used to enhance high-throughput RL training. We assess the influence of latency by testing learning-based methods in a ROS-based environment, where ROS, as an asynchronous system, introduces approximately $45ms$ of delay in node communication.

Tab. 6 details the performance of RL-based methods under the most challenging test, i.e., test 2 of the Multi-Waypoint scenario, with the highest VO, AOL, and the second-highest TO. To further evaluate the baseline's performance at low success rates, we introduce *Progress*, a metric ranging from 0 to 1 that reflects the proportion of the trajectory completed before a collision occurs. The "train w/ latency" column displays results from training with simulated and randomized latencies

between $25ms$ to $50ms$, whereas the "train w/o latency" column serves as a control group. As shown in Tab. 6, "train w latency" significantly improves both success rate and progress by more than $50\%$ for two methods, emphasizing the importance of incorporating randomized latency in more realistic simulation environments and real-world deployments.

**Remark.** *Integrating latency randomization into the training of RL-based methods is essential to enhance real-world applicability.*

## 5 CONCLUSION

We present FlightBench, an open-source benchmark for comparing learning-based and optimization-based methods in ego-vision quadrotor navigation. It includes three optimization-based and two learning-based methods, along with two privileged baselines for thorough comparison. To assess the difficulty of different test configurations, we defined three criteria, resulting in diverse test scenarios. Comprehensive experiments were conducted using these baselines and tests, evaluated by carefully designed performance metrics.

Our results show that while learning-based methods excel in high-speed flight and inference speed, they need improvement in handling complexity and latency robustness. Optimization-based methods remain competitive, producing smooth, flyable trajectories. Analysis of correlations between metrics shows that the proposed task difficulty metrics effectively capture challenges in agility and partial perception. We aim for FlightBench to drive future progress in learning-based navigation for ego-vision quadrotors.

## 6 LIMITATIONS AND FUTURE WORK

FlightBench currently focuses on ego-vision-based quadrotor navigation in static, simulated environments. In the future, we plan to extend its scope to dynamic scenarios and multi-sensor integration, incorporating navigation methods tailored for dynamic environments. This expansion will include comprehensive analysis and a user-friendly hardware platform for real-world validation. Additionally, we will explore sim-to-real transfer techniques to enhance the deployment of navigation methods, with a focus on learning-based approaches.

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

## A IMPLEMENTATION DETAILS

In this section, we detail the implementation specifics of baselines in FlightBench, focusing particularly on methods without open-source code.

### A.1 OPTIMIZATION-BASED METHODS

Fast-Planner (Zhou et al., 2019)[2], EGO-Planner (Zhou et al., 2020)[3], and TGK-Planner (Ye et al., 2020)[4] have all released open-source code. We integrate their open-source code into FlightBench and apply the same set of parameters for evaluation, as detailed in Tab. 7.

Table 7: Key parameters of the optimization-based methods.

|  | Parameter | Value | Parameter | Value |
|---|---|---|---|---|
| All | Max. Vel. | $3.0\,\mathrm{ms^{-1}}$ | Max. Acc. | $6.0\,\mathrm{ms^{-2}}$ |
|  | Obstacle Inflation | 0.09 | Depth Filter Tolerance | 0.15m |
|  | Map Resolution | 0.1m |  |  |
| Fast-Planner EGO-Planner | Max. Jerk | $4.0\,\mathrm{ms^{-3}}$ | Planning Horizon | 6.5 m |
| TGK-Planner | krrt/rho | 0.13 m | Replan Time | 0.005 s |

### A.2 LEARNING-BASED METHODS

Agile (Loquercio et al., 2021)[5] is an open-source learning-based baseline. For each scenario, we finetune the policy from an open-source checkpoint using 100 rollouts before evaluation.

LPA (Song et al., 2023) has not provided open-source code. Therefore, we reproduce the two stage training process based on their paper. The RL training stage involves adding a perception-aware reward to LMT (Penicka et al., 2022) method, which will be introduced in Appendix A.3. At the IL stage, DAgger (Ross et al., 2011) is employed to distill the teacher's experience into an ego-vision student. All our experiments on LPA and LMT use the same set of hyperparameters, as listed in Tab. 8.

### A.3 PRIVILEGED METHODS

SBMT (Penicka & Scaramuzza, 2022)[6] is an open-source sampling-based trajectory generator. Retaining the parameters in their paper, we use SBMT package to generate topological guide path to calculate the task difficulty metrics, and employ PAMPC (Falanga et al., 2018) to track the generated offline trajectories.

We reproduce LMT (Penicka et al., 2022) from scratch based on the original paper, implementing the observation, action, reward function, and training techniques described in the paper. PPO (Yu et al., 2022) is used as the backbone algorithm, and its hyperparameters are listed in Tab. 8.

## B ONBOARD PARAMETERS

The overall system is implemented using ROS 1 Noetic Ninjemys. As mentioned in the main paper, we identify a quadrotor equipped with a depth camera and an IMU from real flight data. The physical characteristics of the quadrotor and its sensors are listed in Tab. 9.

---

[2]`https://github.com/HKUST-Aerial-Robotics/Fast-Planner`
[3]`https://github.com/ZJU-FAST-Lab/ego-planner`
[4]`https://github.com/ZJU-FAST-Lab/TGK-Planner`
[5]`https://github.com/uzh-rpg/agile_autonomy`
[6]`https://github.com/uzh-rpg/sb_min_time_quadrotor_planning`

Table 8: Key hyperparameters used in RL and IL.

| | Parameter | Value | Parameter | Value |
|---|---|---|---|---|
| RL (Yu et al., 2022) | Actor Lr | 5e-4 | Critic Lr | 5e-4 |
| | PPO Epoch | 10 | Batch Size | 51200 |
| | Max Grad. Norm. | 8.0 | Clip Ratio | 0.2 |
| | Entropy Coefficient | 0.01 | | |
| IL (Ross et al., 2011) | Lr | 2e-4 | Training Interval | 20 |
| | Training Epoch | 6 | Max Episode | 2000 |

Table 9: Parameters of the quadrotor and the sensors. $\omega_{xy}$ and $\omega_z$ refer to the angular velocity of roll, pitch, and yaw in the body frame of the quadrotor. "SRT" refers to single rotor thrust and "FOV" denotes the field of view.

| | Parameter | Value | Parameter | Value |
|---|---|---|---|---|
| Quadrotor | Mass | 1.0 kg | Max $\omega_{xy}$ | 8.0 rad/s |
| | Moment of Inertia | [5.9, 6.0, 9.8] g m$^2$ | Max. $\omega_z$ | 3.0 rad/s |
| | Arm Length | 0.125 m | Max SRT | 0.1 N |
| | Torque Constant | 0.0178 m | Min. SRT | 5.0 N |
| Sensors | Depth Range | 4.0 m | Depth FOV | $90° \times 75°$ |
| | Depth Frame Rate | 30 Hz | IMU Rate | 100 Hz |

## C ADDITIONAL RESULTS

### C.1 BENCHMARKING PERFORMANCE

The main paper analyzes the performance of the baselines only on the most challenging tests in each scenario due to space limitations. The full evaluation results are provided in Tab. 10, Tab. 11, and Tab. 12, represented in the form of "mean(std)".

The results indicate that optimization-based methods excel in energy efficiency and trajectory smoothness. In contrast, learning-based approaches tend to adopt more aggressive maneuvers. Although this aggressiveness grants learning-based methods greater agility, it also raises the risk of losing balance in sharp turns.

### C.2 FAILURE CASES

As discussed in Sec. 4.2, our benchmark remains challenging for ego-vision planning methods. In this section, we specifically examine the most demanding tests within the Maze and Multi-Waypoint scenarios to explore how scenarios with high VO and AOL cause failures. Video illustrations of failure cases are provided in the supplementary material.

As shown in Fig. 6(a), Test 3 in the Maze scenario has the highest VO among all tests. Before the quadrotor reaches waypoint 1, its field of view is obstructed by wall (A), making walls (B) and the target space (C) invisible. The sudden appearance of wall (B) often leads to collisions. Additionally, occlusions caused by walls (A) and (C) increase the likelihood of navigating to a local optimum, preventing effective planning towards the target space (C).

Fig. 6(b) illustrates a typical Multi-Waypoint scenario characterized by high VO, TO, and AOL. In this scenario, the quadrotor makes a sharp turn at waypoint 2 while navigating through the waypoints sequentially. The nearest obstacle, column (D), poses a significant challenge due to the need for sudden reactions. Additionally, wall (E), situated close to column (D), often leads to crashes for baselines with limited real-time replanning capabilities.

Table 10: Performance evaluation of different navigation methods in Forest scenario.

| Tests | Metric | Privileged | | | Optimization-based | | Learning-based | |
|---|---|---|---|---|---|---|---|---|
| | | SBMT | LMT | TGK | Fast | EGO | Agile | LPA |
| 1 | Success Rate ↑ | 0.90 | 1.00 | 1.00 | 1.00 | 1.00 | 1.00 | 1.00 |
| | Avg. Spd. ($ms^{-1}$) ↑ | 17.90 (0.022) | 12.10 (0.063) | 2.329 (0.119) | 2.065 (0.223) | 2.492 (0.011) | 3.081 (0.008) | 11.55 (0.254) |
| | Avg. Curv. ($m^{-1}$) ↓ | 0.073 (0.011) | 0.061 (0.002) | 0.098 (0.026) | 0.100 (0.019) | 0.094 (0.010) | 0.325 (0.013) | 0.051 (0.094) |
| | Comp. Time (ms) ↓ | 2.477 (1.350)×10⁵ | 2.721 (0.127) | 11.12 (1.723) | 7.776 (0.259) | 3.268 (0.130) | 5.556 (0.136) | 1.407 (0.036) |
| | Avg. Acc. ($ms^{-3}$) ↓ | 31.54 (0.663) | 9.099 (0.321) | 0.198 (0.070) | 0.254 (0.054) | 54.97 (0.199) | 4.934 (0.385) | 10.89 (0.412) |
| | Avg. Jerk ($ms^{-5}$) ↓ | 4644 (983.6) | 6150 (189.2) | 0.584 (0.216) | 3.462 (1.370) | 3.504 (24.048) | 601.3 (48.63) | 7134 (497.2) |
| 2 | Success Rate ↑ | 0.80 | 1.00 | 1.00 | 1.00 | 1.00 | 1.00 | 1.00 |
| | Avg. Spd. ($ms^{-1}$) ↑ | 14.99 (0.486) | 11.68 (0.072) | 2.300 (0.096) | 2.672 (0.396) | 2.484 (0.008) | 3.059 (0.006) | 9.737 (0.449) |
| | Avg. Curv. ($m^{-1}$) ↓ | 0.069 (0.004) | 0.066 (0.001) | 0.116 (0.028) | 0.068 (0.035) | 0.122 (0.006) | 0.327 (0.025) | 0.071 (0.038) |
| | Comp. Time (ms) ↓ | 2.366 (2.009)×10⁵ | 2.707 (0.079) | 11.75 (1.800) | 7.618 (0.220) | 3.331 (0.035) | 5.541 (0.173) | 1.411 (0.036) |
| | Avg. Acc. ($ms^{-3}$) ↓ | 34.88 (1.224) | 11.14 (0.296) | 0.117 (0.084) | 0.258 (0.148) | 1.265 (0.178) | 4.703 (0.876) | 14.79 (0.564) |
| | Avg. Jerk ($ms^{-5}$) ↓ | 4176 (1654) | 9294 (380.7) | 0.497 (0.484) | 4.017 (1.471) | 83.96 (20.74) | 751.8 (118.4) | 11788 (803.5) |
| 3 | Success Rate ↑ | 0.80 | 1.00 | 0.90 | 0.90 | 1.00 | 0.90 | 1.00 |
| | Avg. Spd. ($ms^{-1}$) ↑ | 15.25 (2.002) | 11.84 (0.015) | 2.300 (0.100) | 2.468 (0.232) | 2.490 (0.006) | 3.058 (0.008) | 8.958 (0.544) |
| | Avg. Curv. ($m^{-1}$) ↓ | 0.065 (0.017) | 0.075 (0.001) | 0.078 (0.035) | 0.059 (0.027) | 0.082 (0.013) | 0.367 (0.014) | 0.080 (0.094) |
| | Comp. Time (ms) ↓ | 2.512 (0.985)×10⁵ | 2.792 (0.168) | 11.54 (1.807) | 7.312 (0.358) | 3.268 (0.188) | 5.614 (0.121) | 1.394 (0.039) |
| | Avg. Acc. ($ms^{-3}$) ↓ | 28.39 (3.497) | 10.29 (0.103) | 0.249 (0.096) | 0.192 (0.119) | 0.825 (0.227) | 4.928 (0.346) | 9.962 (0.593) |
| | Avg. Jerk ($ms^{-5}$) ↓ | 4270 (1378) | 8141 (133.2) | 1.030 (0.609) | 3.978 (1.323) | 58.395 (15.647) | 937.0 (239.2) | 11352 (693.6) |

Table 11: Performance evaluation of different navigation methods in Maze scenario.

| Tests | Metric | Privileged | | Optimization-based | | | Learning-based | |
|---|---|---|---|---|---|---|---|---|
| | | SBMT | LMT | TGK | Fast | EGO | Agile | LPA |
| 1 | Success Rate $\uparrow$ | 0.80 | 1.00 | 0.90 | 1.00 | 0.90 | 1.00 | 0.80 |
| | Avg. Spd. (ms$^{-1}$) $\uparrow$ | 13.66 (1.304) | 10.78 (0.056) | 2.251 (0.123) | 2.097 (0.336) | 2.022 (0.010) | 3.031 (0.004) | 5.390 (0.394) |
| | Avg. Curv. (m$^{-1}$) $\downarrow$ | 0.087 (0.025) | 0.079 (0.002) | 0.154 (0.049) | 0.113 (0.029) | 0.179 (0.011) | 0.135 (0.010) | 0.252 (0.084) |
| | Comp. Time (ms) $\downarrow$ | 1.945 (0.724)$\times 10^5$ | 2.800 (0.146) | 11.76 (0.689) | 7.394 (0.475) | 3.053 (0.035) | 5.535 (0.140) | 1.369 (0.033) |
| | Avg. Acc. (ms$^{-3}$) $\downarrow$ | 34.57 (6.858) | 13.26 (0.374) | 0.277 (0.161) | 0.392 (0.223) | 1.599 (0.220) | 1.023 (0.128) | 18.17 (0.708) |
| | Avg. Jerk (ms$^{-5}$) $\downarrow$ | 4686 (1676) | 10785 (149.1) | 0.809 (0.756) | 3.716 (1.592) | 109.2 (25.81) | 65.94 (15.66) | 8885 (206.4) |
| 2 | Success Rate $\uparrow$ | 0.70 | 1.00 | 0.80 | 0.90 | 0.60 | 0.70 | 0.80 |
| | Avg. Spd. (ms$^{-1}$) $\uparrow$ | 13.67 (0.580) | 10.57 (0.073) | 2.000 (0.065) | 2.055 (0.227) | 2.022 (0.001) | 3.052 (0.003) | 9.314 (0.168) |
| | Avg. Curv. (m$^{-1}$) $\downarrow$ | 0.082 (0.009) | 0.088 (0.002) | 0.157 (0.053) | 0.090 (0.026) | 0.109 (0.002) | 0.193 (0.009) | 0.076 (0.081) |
| | Comp. Time (ms) $\downarrow$ | 2.188 (1.160)$\times 10^5$ | 3.047 (0.196) | 11.58 (0.514) | 7.557 (0.283) | 2.997 (0.022) | 5.579 (0.102) | 1.371 (0.037) |
| | Avg. Acc. (ms$^{-3}$) $\downarrow$ | 31.68 (1.443) | 15.99 (0.274) | 0.252 (0.102) | 0.278 (0.084) | 1.090 (0.147) | 1.772 (0.336) | 10.89 (0.531) |
| | Avg. Jerk (ms$^{-5}$) $\downarrow$ | 2865 (566.8) | 8486 (392.6) | 0.967 (0.759) | 3.999 (1.015) | 103.2 (18.75) | 171.4 (30.66) | 2062 (190.4) |
| 3 | Success Rate $\uparrow$ | 0.60 | 0.90 | 0.50 | 0.60 | 0.20 | 0.50 | 0.50 |
| | Avg. Spd. (ms$^{-1}$) $\uparrow$ | 8.727 (0.168) | 9.616 (0.112) | 1.849 (0.120) | 1.991 (0.134) | 2.189 (0.167) | 2.996 (0.012) | 8.350 (0.286) |
| | Avg. Curv. (m$^{-1}$) $\downarrow$ | 0.313 (0.034) | 0.134 (0.008) | 0.168 (0.060) | 0.229 (0.057) | 0.332 (0.020) | 0.682 (0.084) | 0.214 (0.093) |
| | Comp. Time (ms) $\downarrow$ | 1.649 (1.539)$\times 10^5$ | 2.639 (0.100) | 10.29 (0.614) | 8.918 (0.427) | 3.552 (0.111) | 5.469 (0.081) | 1.422 (0.037) |
| | Avg. Acc. (ms$^{-3}$) $\downarrow$ | 60.73 (5.686) | 26.26 (1.680) | 0.500 (0.096) | 0.786 (0.306) | 1.910 (0.169) | 15.45 (3.368) | 37.30 (1.210) |
| | Avg. Jerk (ms$^{-5}$) $\downarrow$ | 6602 (685.1) | 4649 (305.8) | 6.740 (0.226) | 9.615 (4.517) | 80.54 (7.024) | 2151 (470.2) | 4638 (428.4) |

Table 12: Performance evaluation of different navigation methods in Multi-Waypoint scenario.

| Tests | Metric | Privileged | | | Optimization-based | | Learning-based | |
|---|---|---|---|---|---|---|---|---|
| | | SBMT | LMT | TGK | Fast | EGO | Agile | LPA |
| 1 | Success Rate ↑ | 0.60 | 0.90 | 0.90 | 1.00 | 1.00 | 1.00 | 0.80 |
| | Avg. Spd. $(ms^{-1})$ ↑ | 10.13 (0.343) | 11.06 (0.107) | 1.723 (0.075) | 2.164 (0.140) | 2.512 (0.017) | 3.017 (0.029) | 8.216 (1.459) |
| | Avg. Curv. $(m^{-1})$ → | 0.177 (0.027) | 0.087 (0.004) | 0.119 (0.045) | 0.118 (0.017) | 0.159 (0.012) | 0.406 (0.048) | 0.223 (0.035) |
| | Comp. Time (ms) → | $1.199 \ (0.371) \times 10^5$ | 2.834 (0.163) | 15.32 (0.684) | 9.265 (0.542) | 3.464 (8.786) | 5.610 (0.161) | 1.393 (0.035) |
| | Avg. Acc. $(ms^{-3})$ → | 46.98 (10.46) | 19.67 (1.300) | 0.540 (0.080) | 0.533 (0.169) | 1.063 (0.173) | 7.456 (1.249) | 34.00 (1.551) |
| | Avg. Jerk $(ms^{-5})$ → | 5051 (865.4) | 5641 (358.0) | 6.287 (0.899) | 12.03 (4.145) | 64.50 (12.90) | 1378 (776.3) | 9258 (2238) |
| 2 | Success Rate ↑ | 0.70 | 0.90 | 0.40 | 0.80 | 0.50 | 0.60 | 0.50 |
| | Avg. Spd. $(ms^{-1})$ ↑ | 5.587 (1.351) | 6.880 (0.366) | 1.481 (0.092) | 1.735 (0.241) | 2.132 (0.339) | 3.053 (0.034) | 6.721 (0.980) |
| | Avg. Curv. $(m^{-1})$ → | 0.469 (0.029) | 0.296 (0.031) | 0.463 (0.046) | 0.320 (0.047) | 0.617 (0.216) | 0.668 (0.056) | 0.263 (0.051) |
| | Comp. Time (ms) → | $6.437 \ (0.250) \times 10^5$ | 2.649 (0.185) | 12.32 (2.042) | 9.725 (0.818) | 4.584 (0.734) | 5.683 (0.140) | 1.390 (0.039) |
| | Avg. Acc. $(ms^{-3})$ → | 80.95 (15.10) | 31.23 (1.213) | 1.067 (0.083) | 0.972 (0.470) | 5.060 (1.523) | 16.86 (1.422) | 36.77 (10.36) |
| | Avg. Jerk $(ms^{-5})$ → | 9760 (1000) | 16565 (3269) | 25.52 (7.914) | 22.72 (11.14) | 155.8 (114.1) | 2070 (551.0) | 6187 (956.2) |

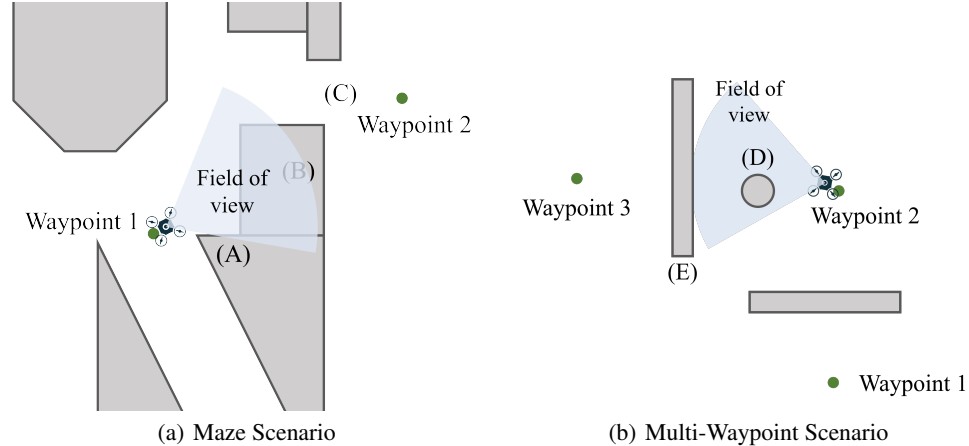

(a) Maze Scenario        (b) Multi-Waypoint Scenario

Figure 6: Visualization of failure cases.

### C.3   ANALYSES ON EFFECTIVENESS OF DIFFERENT METRICS

**Correlation Calculation Method.** As the value of two metrics to be calculated for the correlation coefficient are denoted as $\{x_i\}$, $\{y_i\}$, respectively. The correlation coefficient between $\{x_i\}$ and $\{y_i\}$ defines as

$$\text{Corr}_{x,y} = \frac{\sum_i (x_i - \bar{x})(y_i - \bar{y})}{\sqrt{\sum_i (x_i - \bar{x})^2 \sum (y_i - \bar{y})^2}}, \tag{5}$$

where $\bar{x}$, $\bar{y}$ are the average values of $\{x_i\}$ and $\{y_i\}$.

**Results.** Fig. 7 shows the correlation coefficients between six performance metrics and three difficulty metrics for each method across multiple scenarios. As analyzed in Sec. 4.3, TO and AOL significantly impact the motion performance of two privileged methods (Fig. 7(a) and Fig. 7(b)). In scenarios with high TO and AOL, the baselines tend to fly slower, consume more energy, and exhibit less smoothness. Ego-vision methods are notably influenced by partial perception, making VO a crucial factor. Consequently, high VO greatly decreases the success rate of ego-vision methods, much more so than it does for privileged methods.

When comparing the computation times of different methods, we observe that the time required by learning-based methods primarily depends on the network architecture and is minimally influenced by the scenario. Conversely, the computation times for optimization- and sampling-based methods are affected by both AOL and TO. Scenarios with higher TO and AOL demonstrate increased planning complexity, resulting in longer computation times.

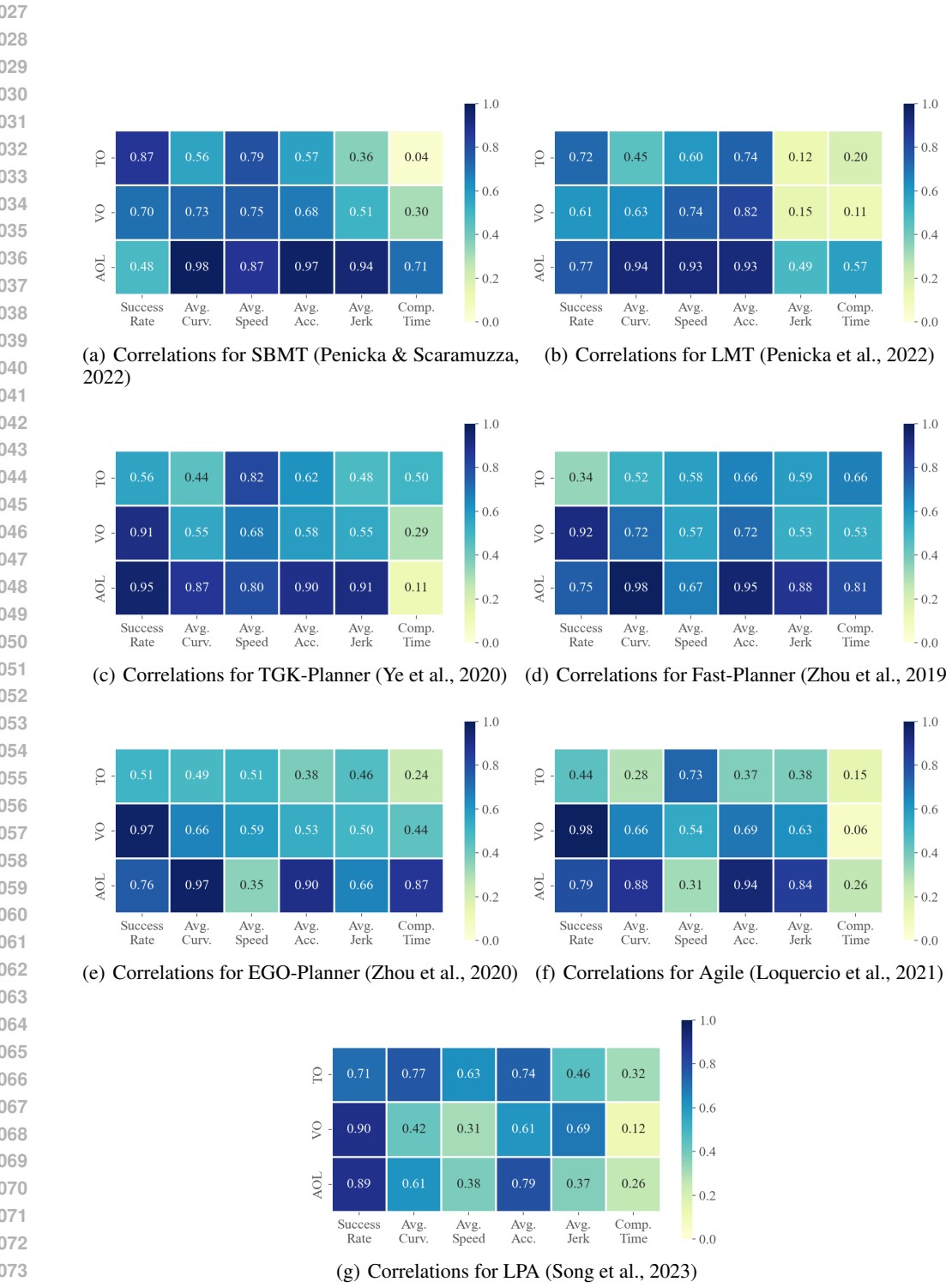

(a) Correlations for SBMT (Penicka & Scaramuzza, 2022)

(b) Correlations for LMT (Penicka et al., 2022)

(c) Correlations for TGK-Planner (Ye et al., 2020)

(d) Correlations for Fast-Planner (Zhou et al., 2019)

(e) Correlations for EGO-Planner (Zhou et al., 2020)

(f) Correlations for Agile (Loquercio et al., 2021)

(g) Correlations for LPA (Song et al., 2023)

Figure 7: Correlation coefficients between three difficulty metrics and evaluation metrics for each method.

