# OpenReview forum: "FlightBench: Benchmarking Learning-based Methods for Ego-vision-based Quadrotors Navigation"
_ICLR.cc/2025/Conference — ICLR 2025 Conference Withdrawn Submission_

### Official Review · Reviewer_uF2v · 2024-11-02

**Soundness:** 3
**Presentation:** 3
**Contribution:** 2
**Rating:** 3
**Confidence:** 4

**Summary:**

This paper introduces a comprehensive benchmark called FlightBench which implements and compares several learning-based methods for ego-vision-based navigation in order to compare them against optimization-based baselines. The paper also develops several assessment metrics, e.g. Traversability Obstruction, View Occlusion, and Angle Over Length.

**Strengths:**

1. This benchmark has capability of 3D scenarios, classical methods and learning methods for planning while allowing sensory inputs in form of vision.
2. Three different scenarios with eight difficulty level has been presented.
3. Performance on different computing platforms have been shown.

**Weaknesses:**

1. The overall paper's core contribution is unclear, i.e. whether it is on the technological side, creating virtual scenarios, etc.
2. It seems this paper has proposed three simulation scenarios with difficulty levels based on different evaluation metrics. However, I don't see any novelty or crucial research contribution from the ICLR perspective. In other words, there are no theoretical or experimental contributions. The paper appears as a system paper where multiple things are simply combined together.
3. Overall, I find this paper a sort of comparison between different planning methods and how they behave in the benchmark. However, It would be interesting to see how this benchmark is advantageous compared to the existing ones.

**Questions:**

See weakness.

---

### Official Review · Reviewer_4m1H · 2024-11-03

**Soundness:** 3
**Presentation:** 3
**Contribution:** 3
**Rating:** 8
**Confidence:** 3

**Summary:**

This submission introduces FlightBench, a benchmark for ego-vision based UAV navigation in simulated environments. The proposed methodology brings together implementations of various traditional (optimisation-based) and learnable solutions from the literature, as well as oracle (environment-aware) baselines, and quantitatively compares them in 3 different environments with varying challenges. Several metrics capturing different aspects of navigation performance are employed (including success rate, speed, latency and other aspects such as energy and dynamic behavior, through proxies), along with a set of metrics to capture task difficulty based on the simulator environment. The manuscript concludes with several insights arising from this comparison, indicating areas for further improvement in the adopted baselines.

**Strengths:**

- The motivation to bring together different methods (learnable and not) for vision-based UAV navigation and compare them head-to-head, addresses an important lack of standardization in the field, that can drive further progress.
- The selected testing environments are quite representative of a wide range of different flying scenarios.
- The evaluation section offers valuable insights on the pros and cons of both categories of UAV navigation solutions, which is very informative and can drive further research and development in the field.

- Overall the paper is well-written and easy to follow, and features useful illustrations. A careful proof reading is needed to correct some sparingly appearing syntax and grammatical errors.

**Weaknesses:**

- The majority of metrics and methods considered in the proposed baseline have been proposed above, or widely used in the community. This may limit the novelty of the proposed approach. However, in my opinion its contribution remains unaffected, as it still provides a comprehensive suite for uniformly evaluating different navigation approaches.
- The proposed benchmark remains rather targeted on static environments and sensing modalities. Driven by the easy extendibility of the proposed framework due to the use of simulated environments, incorporating more sensory cues (e.g. LiDAR, depth, IMU, event camera etc) and more scenarios (e.g. navigation in dynamic environments, drone racing etc) will create a more robust arena and further facilitate the attempt of this work towards standardization in the experiments of the field.
- Relying solely on simulated data, several important aspects of the navigation performance of different methods cannot be accurate evaluated. These include (most crucially) robustness to noisy input data, or lack of scale of training data for deployment in real-world scenarios. At least emulating such cases in the simulated setting will make the obtained results for the proposed benchmark more convincing and representative.


Minor comments:
- Can you define the bounds, or provide qualitative example-score pairs, for the task difficulty metrics defined in Section 3.1 to make the interpretation of Table 2 more intuitive?
- Although the proposed benchmark is more comprehensive, related work could benefit from a broader discussion of other real-world UAV, that can also act as benchmarks for more specific tasks such as ego-motion estimation or drone racing. e.g. :

Delmerico, J., Cieslewski, T., Rebecq, H., Faessler, M. and Scaramuzza, D., 2019, May. Are we ready for autonomous drone racing? the UZH-FPV drone racing dataset. In 2019 International Conference on Robotics and Automation (ICRA) (pp. 6713-6719). IEEE.

**Questions:**

Please consider providing your feedback to the comments raised in the weaknesses section.

---

### Official Review · Reviewer_TSTp · 2024-11-08

**Soundness:** 3
**Presentation:** 3
**Contribution:** 2
**Rating:** 5
**Confidence:** 3

**Summary:**

Summary
The paper introduces a benchmark, named FlightBench for ego-vision based quad rotor navigation methods. In particular, the paper first introduces the paucity of a 3d scene based benchmark to compare learning-based navigation methods against classical optimization methods. To do so, the paper has 3 main contributions:

1. 8 new tasks categorized into 3 scenarios - forests, mazes and multi-waypoints scenarios with varying levels of difficulty, all of which are simulated with Gazebo, Flightmare and ROS.
2. A total of 7 baseline methods categorized into ego-vision based (learning and optimization), and privileged methods.
3. 3 new task difficulty metrics - traversability obstruction, view occlusion and angle over length; to quantify challenges faced during agile navigation.

In addition to comparing learning based methods vs optimization based methods, the paper aims to also analyze navigation performance across different difficulty settings, and the effect of system latency, and provides conclusions on flight quality, flight speed, compute cost, latency and the effectiveness of metrics.

**Strengths:**

1. The paper is generally well-written, and well motivates the need for a 3D scene based benchmark for ego-vision agile navigation, which is currently unaddressed by other benchmarks.
2. The 3 proposed task-difficulty metrics, can comprehensively capture and quantify the challenge of a particular scene with obstacles. In addition, the proposed scenarios are diverse in terms of the task difficulty metrics and capture the different operating conditions often faced.
3. A number of SOTA baselines have been used in the benchmark, covering both learning based, and optimization based methods, as well as methods that leverage additional environmental information. This helps the authors to analyze and remark on various factors in flight performance. Furthermore, the supplementary qualitatively discusses failure cases and their correlation with difficulty metrics.

**Weaknesses:**

1. While the benchmark introduces a variety of scenes, it is still quite limited - as a dataset benchmark, I would have expected more scenarios.
2. Furthermore, the authors do not discuss and analyze the number of scenarios compared to previous baselines. It would be good to have this in the paper. The authors should also discuss the diversity of previous baselines in terms of their proposed task difficulty metrics.
3. Gazebo and ROS-Noetic is used as the simulation platform, which often does not provide realistic scene quality, which could be important for ego-vision based learning. This is also visible through the lack of realistic scene quality. I am curious why the authors did not choose a newer platform like Isaac-Sim and ROS-2 given that they support the new de-facto standards?

Minor comments: This paper seems to be more suitable for a robotics conference/venue than ICLR.

**Questions:**

Pl. refer to weaknesses.

---

### Official Review · Reviewer_g3X4 · 2024-11-13

**Soundness:** 3
**Presentation:** 3
**Contribution:** 2
**Rating:** 3
**Confidence:** 3

**Summary:**

The authors propose FlightBench - a comprehensive benchmark for evaluating ego-vision-based drone navigation methods, comparing learning-based approaches with traditional optimization-based methods. 7 baseline methods are evaluated (2 learning-based, 3 optimization-based, 2 privileged).

The benchmark introduces three key metrics for assessing scenario difficulty: Traversability Obstruction (TO), View Occlusion (VO), and Angle Over Length (AOL).

The test scenarios comprise three categories (Forest, Maze, Multi-Waypoint) with varying difficulty levels.

**Strengths:**

### Novel benchmark
- unified open-source benchmark that enables direct comparison between learning-based and optimization-based methods for UAV navigation
- 3 new quantitative metrics for measuring scenario difficulty
- evaluation across multiple scenarios with varying difficulty levels
### Paper
- well written and illustrated, only minor errors falling through

**Weaknesses:**

### Scope / contribution
- engineering work, the paper appears to be a mix of experiments with existing methods; 3 task difficulty metrics are not enough for ICLR
- out of scope, more suitable for a robotics conference such as ICRA
### Benchmark issues
- limited number of learning-based methods evaluated
- limited to simulated environments

**Questions:**

Apart from the new metrics, how is this work different from Plannie? If I understand correctly, support for real environments is not included.

Otherwise, more metrics could be added (such as battery consumption) and more environmental factors (such as wind) could be also benchmarked for a better real world modelling.

---

### Note · Authors · 2024-11-25

I have read and agree with the venue's withdrawal policy on behalf of myself and my co-authors.